# Learning Disentangled Representation for Robust Person Re-identification

**Chanho Eom**    **Bumsub Ham***
School of Electrical and Electronic Engineering, Yonsei University
`cheom@yonsei.ac.kr`    `bumsub.ham@yonsei.ac.kr`

## Abstract

We address the problem of person re-identification (reID), that is, retrieving person images from a large dataset, given a query image of the person of interest. A key challenge is to learn person representations robust to intra-class variations, as different persons can have the same attribute and the same person's appearance looks different with viewpoint changes. Recent reID methods focus on learning discriminative features but robust to only a particular factor of variations (*e.g.*, human pose), which requires corresponding supervisory signals (*e.g.*, pose annotations). To tackle this problem, we propose to disentangle identity-related and -unrelated features from person images. Identity-related features contain information useful for specifying a particular person (*e.g.*, clothing), while identity-unrelated ones hold other factors (*e.g.*, human pose, scale changes). To this end, we introduce a new generative adversarial network, dubbed *identity shuffle GAN* (IS-GAN), that factorizes these features using identification labels without any auxiliary information. We also propose an identity-shuffling technique to regularize the disentangled features. Experimental results demonstrate the effectiveness of IS-GAN, significantly outperforming the state of the art on standard reID benchmarks including the Market-1501, CUHK03 and DukeMTMC-reID. Our code and models are available online: `https://cvlab-yonsei.github.io/projects/ISGAN/`.

## 1   Introduction

Person re-identification (reID) aims at retrieving person images of the same identity as a query from a large dataset, which is particularly important for finding/tracking missing persons or criminals in a surveillance system. This can be thought of as a *fine-grained* retrieval task in that 1) the data set contains images of the same object class (*i.e.*, person) but with different background clutter and intra-class variations (*e.g.*, pose, scale changes), and 2) they are typically captured with different illumination conditions across multiple cameras possibly with different characteristics and viewpoints. To tackle these problems, reID methods have focused on learning metric space [1, 2, 3, 4, 5, 6] and discriminative person representations [7, 8, 9, 10, 11, 12, 13, 14, 15], robust to intra-class variations and distracting scene details.

Convolutional neural networks (CNNs) have allowed significant advances in person reID in the past few years. Recent methods using CNNs add few more layers for aggregating body parts [9, 10, 11, 12, 16, 17] and/or computing an attention map [13, 14, 15], on the top of *e.g.*, a (cropped) ResNet [18] trained for ImageNet classification [19]. They give state-of-the-art results, but finding person representations robust to various factors is still very challenging. More recent methods exploit generative adversarial networks (GANs) [20] to learn feature representations robust to a particular factor. For example, conditioned on a target pose map and a person image, they generate a new person image of the same identity but with the target pose [21, 22], and the generated image is then used as an additional training data. This allows to learn pose-invariant features, and also has an effect of data augmentation for regularization.

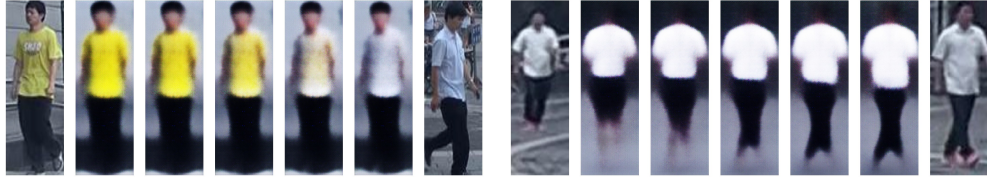

(a) Interpolation between identity-related features     (b) Interpolation between identity-unrelated features

Figure 1: Visual comparison of identity-related and -unrelated features. We generate new person images by interpolating (a) identity-related features and (b) identity-unrelated ones between two images, while fixing the other ones. We can see that identity-related features encode *e.g.*, clothing and color, and identity-unrelated ones involve *e.g.*, human pose and scale changes. Note that we disentangle these features using identification labels only. (Best viewed in color.)

In this paper, we introduce a novel framework, dubbed *identity shuffle GAN* (IS-GAN), that disentangles identity-related and -unrelated features from input person images, without any auxiliary supervisory signals except identification labels. Identity-related features contain information useful for identifying a particular person (*e.g.*, gender, clothing, hair), while identity-unrelated ones hold all other information (*e.g.*, human pose, background clutter, occlusion, scale changes). See Fig. 1 for example. To this end, we propose an identity shuffling technique to disentangle these features using identification labels only within our framework, regularizing the disentangled features. At training time, IS-GAN inputs person images of the same identity and extracts identity-related and -unrelated features. In particular, we divide person images into horizontal parts, and disentangle these features in both image- and part-levels. We then learn to generate new images of the same identity by shuffling identity-related features between the person images. We use the identity-related features only to retrieve person images at test time. We set a new state of the art on standard benchmarks for person reID, and show an extensive experimental analysis with ablation studies.

## 2 Related work

**Person representations.** Recent reID methods provide person representations robust to a particular factor of variations such as human pose, occlusion, and background clutter. Part-based methods [9, 10, 11, 12, 16, 17, 23] represent a person image as a combination of body parts either explicitly or implicitly. Explicit part-based methods use off-the-shelf pose estimators, and extract body parts (*e.g.*, head, torso, legs) with corresponding features [9, 10]. This makes it possible to obtain pose-invariant representations, but off-the-shelf pose estimators often give incorrect pose maps, especially for occluded parts. Instead of using human pose explicitly, a person image is sliced into different horizontal parts of multiple scales in implicit part-based methods [11, 12, 17]. They can exploit various partial information of the image, and provide a feature representation robust to occlusion. Hard [13] or soft [14, 15] attention techniques are also widely exploited in person reID to focus more on discriminative parts while discarding background clutter.

**GAN for person reID.** Recent reID methods leverage GANs to fill the domain gap between source and target datasets [24, 25] or to obtain pose-invariant features [21, 22, 26]. In [24], CycleGAN [27] is used to transform pedestrian images from a source domain to a target one. Similarly, Liu *et al.* [25] use StarGAN [28] to match the camera style of images between source and target domains. Two typical ways of obtaining person representations robust to human pose are to fuse all features extracted from the person images of different poses and to distill pose-relevant information from the images. In [21, 22], new images are generated using GANs conditioned on target pose maps and input person images. Person representations for the generated images are then fused. This approach gives pose-invariant features, but requires auxiliary pose information at test time. It is thus not applicable to new images without pose information. To address this problem, Ge *et al.* [26] introduce FD-GAN that generates a new person image of the same identity as the input with the target pose. Different from the works of [21, 22], it distills identity-related and pose-unrelated features from the input image, getting rid of pose-related information disturbing the reID task. It also does not require additional human pose information during inference.

**Disentangled representations.** Disentangling the factor of variations in CNN features has been widely used to learn the style of a specified factor in order to synthesize new images or extract discriminative features. Mathieu *et al.* [29] introduce a conditional generative model that extracts class-related and -independent features for image retrieval. Liu *et al.* [30] and Bao *et al.* [31]

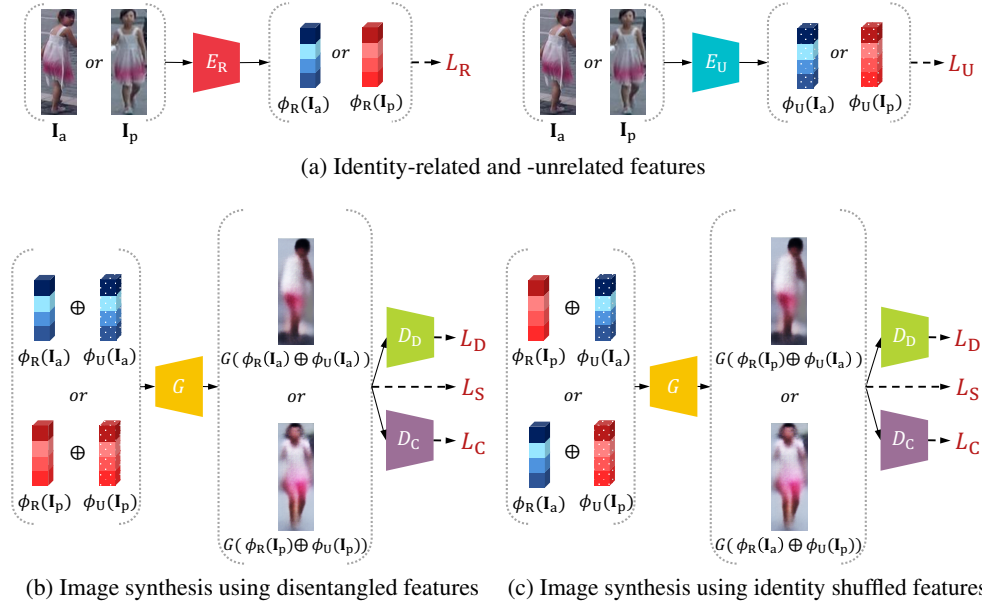

(a) Identity-related and -unrelated features

(b) Image synthesis using disentangled features    (c) Image synthesis using identity shuffled features

Figure 2: Overview of IS-GAN. (a) IS-GAN disentangles identity-related and -unrelated features from person images. (b-c) To regularize the disentanglement process, it learns to generate the same images as the inputs while preserving the identities, using (b) disentangled features and (c) disentangled and identity shuffled ones. We train the encoders, $E_R$ and $E_U$, the generator $G$, the discriminators, $D_D$ and $D_C$, end-to-end. We denote by $\oplus$ a concatenation of features. See text for details.

disentangle the identity and attributes of a face to generate new face images. Denton *et al.* [32] represent videos as stationary and temporally varying components for the prediction of future frames. Unlike these methods, DR-GAN [33] and FD-GAN [26] use a side information (*i.e.*, pose labels) to learn identity-related and pose-unrelated features explicitly for face recognition and person reID, respectively. Other applications of disentangled features include image-to-image translation for producing diverse outputs [34, 35] and domain-specific image deblurring for text restoration [36].

Most similar to ours is FD-GAN [26] that extracts pose-invariant features for person reID. It, however, offers limited feature representations, in that they are not robust to other factors of variations such as scale changes, background clutter and occlusion. Disentangling features with respect to these factors is not feasible within the FD-GAN framework, as this requires corresponding supervisory signals describing the factors (*e.g.*, foreground masks for background clutter). In contrast, IS-GAN factorizes identity-related and -unrelated features without any auxiliary supervisory signals. We also propose to shuffle identity-related features in both image- and part-levels. We empirically find that this is helpful for robust person representations, especially in the case of occlusion and large pose variations that can be seen frequently in person images.

## 3   Approach

We denote by $\mathbf{I}$ and $y \in \{1, 2, ..., C\}$ a person image and an identification label, respectively. $C$ is the number of identities in a dataset. We denote by $\mathbf{I}_a$ and $\mathbf{I}_p$ anchor and positive images, respectively, that share the same identification label. At training time, we input pairs of $\mathbf{I}_a$ and $\mathbf{I}_p$ with the corresponding labels, and train our model to learn identity-related/-unrelated features, $\phi_R(\mathbf{I})$ and $\phi_U(\mathbf{I})$, respectively. At testing time, we compute the Euclidean distance between identity-related features of person images to distinguish whether the identities of them are the same or not.

### 3.1   Overview

IS-GAN mainly consists of five components (Fig. 2): An identity-related encoder $E_R$, an identity-unrelated encoder $E_U$, a generator $G$, a domain discriminator $D_D$, and a class discriminator $D_C$. Given pairs of $\mathbf{I}_a$ and $\mathbf{I}_p$, the encoders, $E_R$ and $E_U$, learn identity-related features, $\phi_R(\mathbf{I}_a)$ and $\phi_R(\mathbf{I}_p)$, and identity-unrelated ones, $\phi_U(\mathbf{I}_a)$ and $\phi_U(\mathbf{I}_p)$, respectively (Fig. 2(a)). To encourage identity-related and -unrelated encoders to disentangle these features from the input images, we

train the generator $G$, such that it synthesizes the same images as $\mathbf{I}_\mathrm{a}$ from $\phi_\mathrm{R}(\mathbf{I}_\mathrm{a}) \oplus \phi_\mathrm{U}(\mathbf{I}_\mathrm{a})$ and $\phi_\mathrm{R}(\mathbf{I}_\mathrm{p}) \oplus \phi_\mathrm{U}(\mathbf{I}_\mathrm{a})$, where we denote by $\oplus$ a concatenation of features (Fig. 2(b-c)). Similarly, it generates the same images as $\mathbf{I}_\mathrm{p}$ from $\phi_\mathrm{R}(\mathbf{I}_\mathrm{p}) \oplus \phi_\mathrm{U}(\mathbf{I}_\mathrm{p})$ and $\phi_\mathrm{R}(\mathbf{I}_\mathrm{a}) \oplus \phi_\mathrm{U}(\mathbf{I}_\mathrm{p})$. Since $\mathbf{I}_\mathrm{a}$ and $\mathbf{I}_\mathrm{p}$ have the same identity but with *e.g.* different poses, scales, and illumination, this *identity shuffling* encourages the identity-related encoder $E_\mathrm{R}$ to extract features robust to such variations, focusing on the shared information between $\mathbf{I}_\mathrm{a}$ and $\mathbf{I}_\mathrm{p}$, while enforcing the identity-unrelated encoder $E_\mathrm{U}$ to capture other factors. We also perform the feature disentanglement and identity shuffling in a part-level by dividing the input images into multiple horizontal regions (Fig. 3). Given the generated images, the class discriminator $D_\mathrm{C}$ determines their identification labels as either that of $\mathbf{I}_\mathrm{a}$ or $\mathbf{I}_\mathrm{p}$, and the domain discriminator $D_\mathrm{D}$ tries to distinguish real and fake images. IS-GAN is trained end-to-end using identification labels without any auxiliary supervision.

## 3.2 Baseline model

We exploit a network architecture similar to [12] for the encoder $E_\mathrm{R}$. It has three branches on top of a backbone network, where each branch has the same structure but different parameters. We call them as part-1, part-2, and part-3 branches, that slice a feature map from the network equally into one, two, and three horizontal regions, respectively. The part-1 branch provides a global feature of the entire person image. Other branches give both global and local features describing body parts, where the local features are extracted from corresponding horizontal regions. For example, the part-3 branch outputs three local features and a single global one. Accordingly, we extract $K$ features from the encoder $E_\mathrm{R}$ in total, where $K = 8$ in our case. Without loss of generality, we can use additional branches to consider different horizontal regions of multiple scales.

**ID loss.** We denote by $\mathbf{I}^k$ and $\phi_\mathrm{R}^k$ $(k = 1 \dots K)$ horizontal regions of multiple scales and corresponding embedding functions that extract identity-related features, respectively. Following other reID methods [11, 12, 15, 22], we formulate the reID problem as a multi-class classification task, and train the encoder $E_\mathrm{R}$ with a cross-entropy loss. Concretely, a loss function $\mathcal{L}_\mathrm{R}$ to learn the embedding function $\phi_\mathrm{R}^k$ is defined as follows:

$$\mathcal{L}_\mathrm{R} = -\sum_{c=1}^{C} \sum_{k=1}^{K} q_c^k \log p(c | \mathbf{w}_c^k \phi_\mathrm{R}^k(\mathbf{I}^k)), \tag{1}$$

where $\mathbf{w}_c^k$ is the classifier parameters associated with the identification label $c$ and the region $\mathbf{I}^k$. $q_c^k$ is the index label with $q_c^k = 1$ if the label $c$ corresponds to the identity of the image $\mathbf{I}^k$ (*i.e.*, $c = y$) and $q_c^k = 0$ otherwise. The probability of $\mathbf{I}^k$ with the label $c$ is defined using a softmax function as

$$p(c | \mathbf{w}_c^k \phi_\mathrm{R}^k(\mathbf{I}^k)) = \frac{\exp(\mathbf{w}_c^k \phi_\mathrm{R}^k(\mathbf{I}^k))}{\sum_{i=1}^{C} \exp(\mathbf{w}_c^k \phi_\mathrm{R}^k(\mathbf{I}^k))}. \tag{2}$$

We concatenate all features from three branches, and use it as an identity-related feature $\phi_\mathrm{R}(\mathbf{I})$ for the image $\mathbf{I}$, that is, $\phi_\mathrm{R}(\mathbf{I}) = \phi_\mathrm{R}^1(\mathbf{I}^1) \oplus ... \oplus \phi_\mathrm{R}^K(\mathbf{I}^K)$.

## 3.3 IS-GAN

The identity-related feature $\phi_\mathrm{R}(\mathbf{I})$ from the encoder $E_\mathrm{R}$ contains information useful for person reID, such as clothing, texture, and gender. However, the feature $\phi_\mathrm{R}(\mathbf{I})$ learned using the classification loss in (1) only may have other information that is not related to or even distracts specifying a person (*e.g.*, human pose, background clutter, scale), and thus it is not enough to handle these factors of variations. To address this problem, we use an additional encoder $E_\mathrm{U}$ to extract the identity-unrelated feature $\phi_\mathrm{U}(\mathbf{I})$, and train the encoders such that they give disentangled feature representations for identifying a person. The key idea behind the feature disentanglement is to *distill* identity-unrelated information from the identity-related feature, and vice versa. To this end, we propose to leverage image synthesis using an identity shuffling technique. Applying this to the whole body and its parts regularizes the disentangled features. Two discriminators allow to generate realistic person images of particular identities, further regularizing the disentanglement process.

**Identity-shuffling loss.** We assume that the disentangled person representation satisfies the following conditions: 1) An original image should be reconstructed from its identity-related and -unrelated features; 2) The shared information between different images of the same identity corresponds to the identity-related feature. To implement this, the generator $G$ is required to reconstruct an anchor image $\mathbf{I}_\mathrm{a}$ from $\phi_\mathrm{R}(\mathbf{I}_\mathrm{a}) \oplus \phi_\mathrm{U}(\mathbf{I}_\mathrm{a})$ and $\phi_\mathrm{R}(\mathbf{I}_\mathrm{p}) \oplus \phi_\mathrm{U}(\mathbf{I}_\mathrm{a})$ while synthesizing a positive image $\mathbf{I}_\mathrm{p}$ from

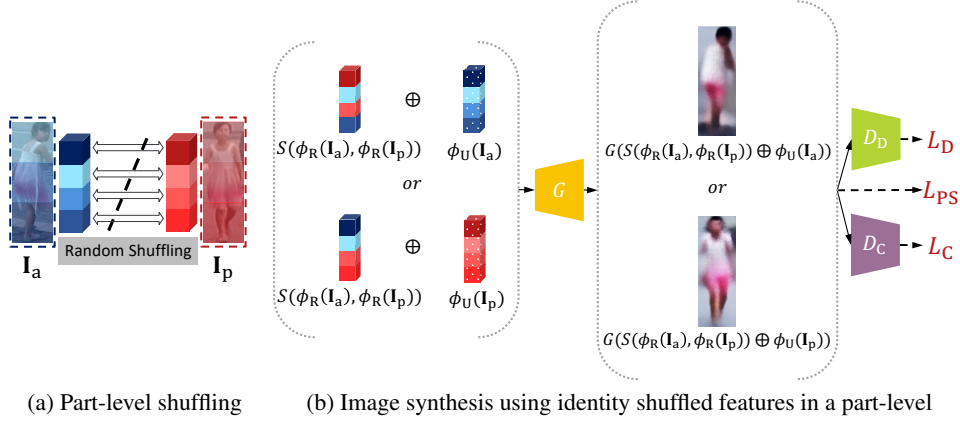

(a) Part-level shuffling    (b) Image synthesis using identity shuffled features in a part-level

Figure 3: (a) We randomly swap local features between anchor and positive images. (b) Similar to Fig. 2(c), we generate person images with identity-related features but shuffled in a part-level and identity-unrelated ones. See text for details.

$\phi_{\mathrm{R}}(\mathbf{I}_{\mathrm{p}}) \oplus \phi_{\mathrm{U}}(\mathbf{I}_{\mathrm{p}})$ and $\phi_{\mathrm{R}}(\mathbf{I}_{\mathrm{a}}) \oplus \phi_{\mathrm{U}}(\mathbf{I}_{\mathrm{p}})$ (Fig. 2(b-c)). We define an identity-shuffling loss as follows:

$$\mathcal{L}_{\mathrm{S}} = \sum_{i,j \in \{\mathrm{a,p}\}} \|\mathbf{I}_i - G(\phi_{\mathrm{R}}(\mathbf{I}_j) \oplus \phi_{\mathrm{U}}(\mathbf{I}_i))\|_1. \tag{3}$$

The generator acts as an auto-encoder when $i = j$, enforcing the combination of identity-related and -unrelated features from the same image to contain all information in order to reconstruct the original image. When $i \neq j$, it encourages the encoder $E_{\mathrm{R}}$ to extract the same identity-related features, $\phi_{\mathrm{R}}(\mathbf{I}_{\mathrm{a}})$ and $\phi_{\mathrm{R}}(\mathbf{I}_{\mathrm{p}})$ from a pair of $\mathbf{I}_{\mathrm{a}}$ and $\mathbf{I}_{\mathrm{p}}$, focusing on the consistent information between them. Other factors, not shared by $\mathbf{I}_{\mathrm{a}}$ and $\mathbf{I}_{\mathrm{p}}$, are encoded into the identity-unrelated features, $\phi_{\mathrm{U}}(\mathbf{I}_{\mathrm{a}})$ and $\phi_{\mathrm{U}}(\mathbf{I}_{\mathrm{p}})$.

**Part-level shuffling loss.** We also apply the identity shuffling technique to part-level features (Fig. 3). We randomly choose local features from $\phi_{\mathrm{R}}(\mathbf{I}_{\mathrm{a}})$, and swap them with corresponding ones from $\phi_{\mathrm{R}}(\mathbf{I}_{\mathrm{p}})$ at the same locations, and vice versa (Fig. 3(a)). This assumes that horizontal regions in a person image contain discriminative body parts sufficient for distinguishing its identity. Similar to (3), we compute the discrepancy between the original image and its reconstruction from the identity-related features shuffled in a part-level and the identity-unrelated ones (Fig. 3(b)), and define a part-level shuffling loss as

$$\mathcal{L}_{\mathrm{PS}} = \sum_{\substack{i,j \in \{\mathrm{a,p}\} \\ i \neq j}} \|\mathbf{I}_i - G(S(\phi_{\mathrm{R}}(\mathbf{I}_i), \phi_{\mathrm{R}}(\mathbf{I}_j)) \oplus \phi_{\mathrm{U}}(\mathbf{I}_i))\|_1, \tag{4}$$

where we denote by $S$ a region-wise shuffling operator. The part-level identity shuffling has the following advantages: 1) It enables our model to see various combinations of identity-related features for individual body parts, regularizing a feature disentanglement process; 2) It imposes feature consistency between corresponding parts of the images.

**KL divergence loss.** We disentangle the identity-related and -unrelated features using identification labels only. Although we train the encoders separately to extract these features, where they share a backbone network with different heads, the generator $G$ may largely rely on the identity-unrelated features to synthesize new person images in (3) and (4), while ignoring the identity-related ones, which distracts the feature disentanglement process. To circumvent this issue, we encourage the identity-unrelated features to have the normal distribution $\mathcal{N}(0, 1)$ with zero mean and unit variance, and formulate this using a KL divergence loss as follows:

$$\mathcal{L}_{\mathrm{U}} = \sum_{k=1}^{K} D_{\mathrm{KL}}\big(\phi_{\mathrm{U}}^k(\mathbf{I}^k)||\mathcal{N}(0, 1)\big) \tag{5}$$

where $D_{KL}(p||q) = -\int p(z) log \frac{p(z)}{q(z)}$. The KL divergence loss regularizes the identity-unrelated features by limiting the distribution range, such that they do not contain much identity-related

information [31, 35, 36]. This enforces the generator $G$ to use the identity-related features when synthesizing new person images, facilitating the disentanglement process.

**Domain and class losses.** To train the generator $G$ in (3) and (4), we use two discriminators $D_\mathrm{D}$ and $D_\mathrm{C}$. The domain discriminator $D_\mathrm{D}$ [20] helps the generator $G$ to synthesize more realistic person images, and the class discriminator $D_\mathrm{C}$ [37] encourages the synthesized images to have the identification labels of anchor and positive images, further regularizing the feature learning process. Concretely, we define a domain loss $\mathcal{L}_\mathrm{D}$ as

$$\mathcal{L}_\mathrm{D} = \max_{D_\mathrm{D}} \sum_{i \in \{a,p\}} \log D_\mathrm{D}(\mathbf{I}_i) + \sum_{i,j \in \{a,p\}} \log(1 - D_\mathrm{D}(G(\phi_\mathrm{R}(\mathbf{I}_j) \oplus \phi_\mathrm{U}(\mathbf{I}_i)))) \qquad (6)$$
$$+ \sum_{\substack{i,j \in \{a,p\} \\ i \neq j}} \log(1 - D_\mathrm{D}(G(S(\phi_\mathrm{R}(\mathbf{I}_i), \phi_\mathrm{R}(\mathbf{I}_j)) \oplus \phi_\mathrm{U}(\mathbf{I}_i)))).$$

The domain discriminator $D_D$ is trained, such that it distinguishes real and fake images while the generator $G$ tries to synthesize more realistic images to fool $D_D$. A class loss $\mathcal{L}_\mathrm{C}$ is defined as

$$\mathcal{L}_\mathrm{C} = - \sum_{i \in \{a,p\}} \log D_\mathrm{C}(\mathbf{I}_i) - \sum_{i,j \in \{a,p\}} \log(D_\mathrm{C}(G(\phi_\mathrm{R}(\mathbf{I}_j) \oplus \phi_\mathrm{U}(\mathbf{I}_i)))) \qquad (7)$$
$$- \sum_{\substack{i,j \in \{a,p\} \\ i \neq j}} \log(D_\mathrm{C}(G(S(\phi_\mathrm{R}(\mathbf{I}_i), \phi_\mathrm{R}(\mathbf{I}_j)) \oplus \phi_\mathrm{U}(\mathbf{I}_i)))).$$

The class discriminator $D_\mathrm{C}$ classifies the identification labels of generated and input person images. When the generator $G$ synthesizes a hard-to-classify image without sufficient identity-related information, the class discriminator $D_\mathrm{C}$ would be confused to determine the identification label of the generated image. The generator $G$ thus tries to synthesize a person image of the particular identity associated with the identity-related features, $\phi_\mathrm{R}(\mathbf{I}_j)$ and $S(\phi_\mathrm{R}(\mathbf{I}_i), \phi_\mathrm{R}(\mathbf{I}_j))$.

**Training loss.** The overall objective is a weighted sum of all loss functions defined as:

$$\mathcal{L}(E_\mathrm{R}, E_\mathrm{U}, G, D_\mathrm{D}, D_\mathrm{C}) = \lambda_\mathrm{R}\mathcal{L}_\mathrm{R} + \lambda_\mathrm{U}\mathcal{L}_\mathrm{U} + \lambda_\mathrm{S}\mathcal{L}_\mathrm{S} + \lambda_\mathrm{PS}\mathcal{L}_\mathrm{PS} + \lambda_\mathrm{D}\mathcal{L}_\mathrm{D} + \lambda_\mathrm{C}\mathcal{L}_\mathrm{C}, \qquad (8)$$

where $\lambda_\mathrm{R}, \lambda_\mathrm{U}, \lambda_\mathrm{S}, \lambda_\mathrm{PS}, \lambda_\mathrm{D}, \lambda_\mathrm{C}$ are the weighting factors for each loss.

## 4 Experiments

### 4.1 Implementation details

**Network architecture.** We exploit a ResNet-50 [18] trained for ImageNet classification [19]. Specifically, we use the network cropped at `conv4-1` as our backbone to extract CNN features. On top of that, we add two heads for the identity-related and -unrelated encoders. Each encoder has part-1, part-2, and part-3 branches that consist of two convolutional, global max pooling, and bottleneck layers but with different number of channels and network parameters. The part-1, part-2, and part-3 branches in the encoders give feature maps of size $1 \times 1 \times p$, $1 \times 1 \times 3p$, and $1 \times 1 \times 4p$, respectively. See Section 3.2 for details. We set the size of $p$ (*i.e.*, the number of channels) to 256 and 64 for the identity-related and -unrelated encoders, respectively. We concatenate all features from three branches for each encoder, and obtain the identity-related and -unrelated features. The generator consists of a series of six transposed convolutional layers with batch normalization [38], Leaky ReLU [39] and Dropout [40]. It inputs identity-related and -unrelated features, a noise vector, and a one-hot vector encoding an identification label whose dimensions are 2048, 512, 128 and $C$, respectively. The domain and class discriminators share five blocks consisting of a convolutional layer with stride 2 with instance normalization [41] and Leaky ReLU [39], but have different heads. For the domain discriminator, we add two more blocks, resulting in a features map of size $12 \times 4$. We then use this as an input to PatchGAN [42]. For the class discriminator, we add one more block followed by a fully connected layer.

**Dataset and evaluation metric.** We compare our model to the state of the art on person reID with the following benchmark datasets: Market-1501 [43], CUHK03 [44] and DukeMTMC-reID [45]. The Market-1501 dataset [43] contains $1,501$ pedestrian images captured from six viewpoints. Following the standard split [43], we use $12,936$ images of $751$ identities for training and $19,732$ images of

Table 1: Quantitative comparison with the state of the art on Market-1501 [43], CUHK03 [44] and DukeMTMC-reID [45] in terms of rank-1 accuracy(%) and mAP(%). Numbers in bold indicate the best performance and underscored ones are the second best. †: ReID methods trained using both classification and (hard) triplet losses; ∗: Our implementation.

| Methods | f-dim | Market-1501 | | CUHK03 | | | | DukeMTMC-reID | |
| | | | | labeled | | detected | | | |
| | | R-1 | mAP | R-1 | mAP | R-1 | mAP | R-1 | mAP |
| --- | --- | --- | --- | --- | --- | --- | --- | --- | --- |
| IDE [51] | 2,048 | 73.9 | 47.8 | 22.2 | 21.0 | 21.3 | 19.7 | - | - |
| SVDNet [52] | 2,048 | 82.3 | 62.1 | 40.9 | 37.8 | 41.5 | 37.3 | 76.7 | 56.8 |
| DaRe† [53] | 128 | 86.4 | 69.3 | 58.1 | 53.7 | 55.1 | 51.3 | 75.2 | 57.4 |
| PN-GAN [21] | 1,024 | 89.4 | 72.6 | - | - | - | - | 73.6 | 53.2 |
| MLFN [54] | 1,024 | 90.0 | 74.3 | 54.7 | 49.2 | 52.8 | 47.8 | 81.0 | 62.8 |
| FD-GAN [26] | 2,048 | 90.5 | 77.7 | - | - | - | - | 80.0 | 64.5 |
| HA-CNN [15] | 1,024 | 91.2 | 75.7 | 44.4 | 41.0 | 41.7 | 38.6 | 80.5 | 63.8 |
| Part-Aligned† [23] | 512 | 91.7 | 79.6 | - | - | - | - | 84.4 | 69.3 |
| PCB [11] | 12,288 | 92.3 | 77.4 | - | - | 59.7 | 53.2 | 81.7 | 66.1 |
| PCB+RPP [11] | 12,288 | 93.8 | 81.6 | - | - | 62.8 | 56.7 | 83.3 | 69.2 |
| HPM [49] | 3,840 | 94.2 | 82.7 | - | - | 63.9 | 57.5 | 86.6 | 74.3 |
| DG-Net [55] | 1,024 | 94.8 | 86.0 | - | - | - | - | 86.6 | 74.8 |
| MGN† [12] | 2,048 | **95.7** | 86.9 | 68.0 | 67.4 | 66.8 | 66.0 | **88.7** | 78.4 |
| MGN†,∗ [12] | 2,048 | 94.5 | 84.8 | 69.2 | 67.6 | 65.7 | 62.1 | 88.2 | 76.7 |
| IS-GAN | 2,048 | 95.2 | **87.1** | **74.1** | **72.5** | **72.3** | **68.8** | **90.0** | **79.5** |

750 identities for testing. The CUHK03 dataset [44] contains $14,096$ images of $1,467$ identities captured by two cameras. For the training/testing split, we follow the experimental protocol in [46]. The DukeMTMC-reID dataset [45], a subset of the DukeMTMC [47], provides $36,411$ images of $1,812$ identities captured by eight cameras, including $408$ identities (distractor IDs) that appear in only one camera. We use the training/test split provided by [45] corresponding $16,522$ images of 702 identities for training and $2,228$ query and $17,661$ gallery images of 702 identities for testing. We measure mean average precision (mAP) and cumulative matching characteristics (CMC) at rank-1 for evaluation.

**Training.** To train the encoders and the generator, we use the Adam [48] optimizer with $\beta_1 = 0.9$ and $\beta_2 = 0.999$. For the discriminators, we use the stochastic gradient descent with momentum of $0.9$. Similar to the training scheme in [26], we train IS-GAN in three stages: In the first stage, we train the identity-related encoder $E_R$ using the loss function $\mathcal{L}_R$, which corresponds to the baseline model, for 300 epochs over the training data. A learning rate is set to 2e-4. In the second stage, we fix the baseline, and train the identity-unrelated encoder $E_U$, the generator $G$, and the discriminators $D_D$ and $D_C$ with the corresponding losses $\mathcal{L}_U$, $\mathcal{L}_S$, $\mathcal{L}_{PS}$, $\mathcal{L}_D$, and $\mathcal{L}_C$. This process iterates for 200 epochs with the learning rate of 2e-4. Finally, we train the whole network end-to-end with the learning rate of 2e-5 for 100 epochs. Following [49], we resize all image into $384 \times 128$. We augment the datasets with horizontal flipping and random erasing [50]. Note that random erasing is used only in the first stage, as we empirically find that it hinders the disentanglement process. For mini-batch, we randomly select 4 different identities, and sample a set of 4 images for each identity.

**Hyperparameter.** We empirically find that training with a large value of $\lambda_U$ is unstable. We thus set $\lambda_U$ to 0.001 in the second stage, and increase it to 0.01 in the third stage to regularize the disentanglement. Following [26, 35], we fix $\lambda_S$ and $\lambda_D$ to 10 and 1, respectively. To set other parameters, we randomly split IDs in the training dataset of Market-1501 [43] into 651/100 and used corresponding images as training/validation sets. We use a grid search to set the parameters ($\lambda_R = 20$, $\lambda_{PS} = 10$, $\lambda_C = 2$) with $\lambda_R \in \{5, 10, 20\}$, $\lambda_{PS} \in \{5, 10, 20\}$, and $\lambda_C \in \{1, 2\}$ on the validation split. We fix all parameters and train our models on Market-1501 [43], CUHK03 [44] and DukeMTMC-reID [45].

## 4.2 Results

**Quantitative Comparison with the state of the art.** We show in Table 1 rank-1 accuracy and mAP for Market-1501 [43], CUHK03 [44] and DukeMTMC-reID [45], and compare IS-GAN with the state of the art including FD-GAN [26], PCB+RPP [11], DG-Net [55], and MGN [12]. We

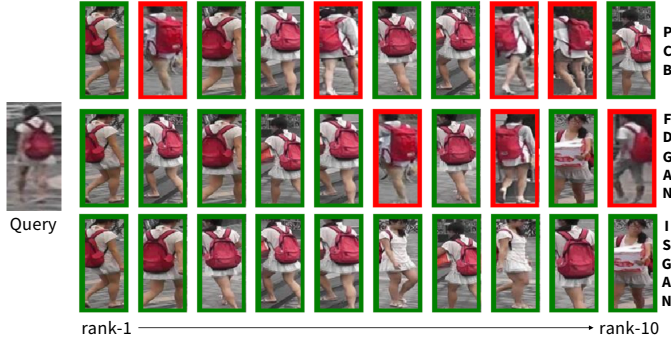
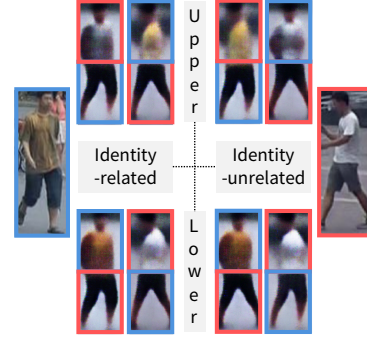

Figure 4: Visual comparison of retrieval results on Market-1501 [43]. Results with green boxes have the same identity as the query, while those with red boxes do not. (Best viewed in color.)

Figure 5: An example of generated images using a part-level identity shuffling technique. (Best viewed in color.)

use a single query, and do not use any post-processing techniques (*e.g.*, a re-ranking method [46]). We achieve 95.2% rank-1 accuracy and 87.1% mAP on Market-1501 [43], 74.1%/72.3% rank-1 accuracy and 72.5%/68.8% mAP with labeled/detected images on CUHK03 [44], and 90.0% rank-1 accuracy and 79.5% mAP on DukeMTMC-reID [45], setting a new state of the art on CUHK03 and DukeMTMC-reID. Note that IS-GAN is the first model we are aware of that achieves more 90% rank-1 accuracy on DukeMTMC-reID [45].

FD-GAN [26] is similar to IS-GAN in that both use a GAN-based distillation technique for person reID. It extracts identity-related and pose-unrelated features using extra pose labels. Distilling other factors except for human pose is not feasible. IS-GAN on the other hand disentangles identity-related and -unrelated features through identity shuffling, factorizing other factors irrelevant to person reID, such as pose, scale, background clutter, without supervisory signals for them. Accordingly, the identity-related feature of IS-GAN is much more robust to such factors of variations than the identity-related and pose-unrelated one of FD-GAN, showing the better performance on Market-1501 and DukeMTMC-reID. Note that the results of FD-GAN on CUHK03 are excluded, as it uses a different training/test split.

DG-Net [55] also use a feature distillation technique, but appearance/structure features in DG-Net are completely different from identity-related/-unrelated ones in IS-GAN. DG-Net computes the features by AdaIN [56], widely used in image stylization, and thus they contain style/content information, rather than identity-related/-unrelated one. Figure 9 in Appendix of [55] visualizes generated person images when structure features (analogous to identity-unrelated features of IS-GAN) are changed only. We can see that DG-Net even changes the entire attributes (*e.g.*, gender) except the color information, suggesting that the structure features also contain identity-related cues. As a result, IS-GAN outperforms DG-Net for all benchmarks by a large margin.

MGN [12] uses the same backbone network as IS-GAN to extract initial part-level features. As it is trained with a hard-triplet loss, the part-level features of MGN capture discriminative attributes of person images well. For Market-1501, MGN shows the reID performance comparable with IS-GAN, and performs slightly better in terms of rank-1 accuracy. Note that, compared to other datasets, it contains person images of less pose and attribute variations. The reID performance of MGN, however, drops significantly on other datasets, especially for CUHK03, where the same person is captured with different poses, viewpoints, background, and occlusion, demonstrating that the person representations for MGN are not robust to such factors of variations.

**Qualitative Comparison with the state of the art.** Figure 4 shows person retrieval results of PCB [11], FD-GAN [26], and ours on Market-1501 [43]. We can see that PCB mainly focuses on clothing color, retrieving many person images of different identities from the query. FD-GAN using the identity-related and pose-unrelated features shows the robustness to pose variations. It, however, largely relies on color information. For example, FD-GAN even retrieves person images of different genders, just because the persons carry a red bag and put on a white top. In contrast, IS-GAN retrieves person images of the same identity as the query correctly. We can see that identity-related features in IS-GAN are robust to large pose variations, occlusion, background clutter, and scale changes.

Table 2: Ablation studies of IS-GAN on Market-1501 [43], CUHK03 [44] and DukeMTMC-reID [45] in terms of rank-1 accuracy(%) and mAP(%).

| | Losses | | | | | | Market-1501 | | CUHK03-labeled | | DukeMTMC-reID | |
|---|---|---|---|---|---|---|---|---|---|---|---|---|
| | $\mathcal{L}_\text{R}$ | $\mathcal{L}_\text{U}$ | $\mathcal{L}_\text{S}$ | $\mathcal{L}_\text{PS}$ | $\mathcal{L}_\text{D}$ | $\mathcal{L}_\text{C}$ | R-1 | mAP | R-1 | mAP | R-1 | mAP |
| Baseline | ✓ | | | | | | 93.9 | 84.1 | 68.4 | 65.9 | 86.6 | 74.9 |
| IS-GAN | ✓ | ✓ | ✓ | | | | 94.8 | 87.0 | 73.4 | 72.3 | 89.5 | **79.5** |
| | ✓ | ✓ | ✓ | ✓ | | | 95.0 | **87.1** | 73.9 | 72.1 | 89.7 | 79.4 |
| | ✓ | ✓ | ✓ | ✓ | ✓ | | 94.9 | 87.0 | 73.9 | 72.3 | 89.8 | 79.4 |
| | ✓ | ✓ | ✓ | ✓ | | ✓ | 95.1 | 86.9 | 73.7 | 72.3 | 89.7 | **79.5** |
| | ✓ | ✓ | ✓ | ✓ | ✓ | ✓ | **95.2** | **87.1** | **74.1** | **72.5** | **90.0** | **79.5** |

**Ablation study.** We show an ablation analysis on different losses in IS-GAN. We measure rank-1 accuracy and mAP, and report results on Market-1501 [43], CUHK03 [44] and DukeMTMC-reID [45] in Table 2. From the first and second rows, we can clearly see that disentangling identity-related and -unrelated features using an identity shuffling technique gives better results on all datasets, but the performance gain for the CUHK03 [44], which typically contains person images of large pose variations and similar attributes, is more significant. The third row shows that applying the identity shuffling technique in a part-level further boosts the reID performance. The last three rows demonstrate that domain and class discriminators are complementary, and combining all losses gives the best results.

**Part-level shuffling loss.** We show in Table 3 the effect of the part-level shuffling loss for different numbers of body parts. We can see that 1) the part-level shuffling loss generalizes well across different numbers of body parts, and 2) IS-GAN shows better performance as more body parts are used. To further evaluate the generalization ability of our model, we use PCB [11] as our baseline and add IS-GAN on top of that. We modify the network architecture such that each part-level feature has the size of $1 \times 1 \times 256$ for an efficient computation. Note that the original PCB also gives six part-level features, but with the size of $1 \times 1 \times 2,048$. The rank-1/mAP results of PCB, PCB+IS-GAN (w/o $\mathcal{L}_\text{PS}$), and PCB+IS-GAN are 91.0/74.2, 92.1/78.3, and 92.6/78.5, respectively, showing that our model improves the performance of PCB consistently.

Table 3: Ablation studies of different numbers of body parts on Market-1501 [43].

| | $\mathcal{L}_\text{PS}$ | R-1 | mAP |
|---|---|---|---|
| part-2 | X | 93.6 | 82.6 |
| | ✓ | 93.9 | 82.9 |
| part-3 | X | 94.1 | 82.9 |
| | ✓ | 94.4 | 83.0 |
| part-1,2 | X | 94.4 | 84.4 |
| | ✓ | 94.7 | 84.5 |
| part-1,3 | X | 94.5 | 84.9 |
| | ✓ | 94.7 | 85.2 |

**Visual analysis for disentangled features.** Figure 5 visualizes the ability of IS-GAN to disentangle identity-related and -unrelated features in a part-level. We show an example of generated images using a part-level identity shuffling technique. Specifically, we shuffle the identity-related/-unrelated features for upper/lower parts between person images of different identities. When identity-related features are shuffled *e.g.*, in the upper left picture, we can see that IS-GAN changes colors of T-shirts between persons but with the same pose and background. This suggests that the identity-related features do not contain pose and background information. Interestingly, when identity-unrelated features are shuffled, IS-GAN generates new images where background and pose information for the corresponding parts are changed. For example in the upper right picture, the person looking at the front side now sees the left side and vice versa when shuffling the features between upper parts, while preserving the shapes of the legs in the lower parts.

## 5 Conclusion

We have presented a novel framework, IS-GAN, to learn disentangled representations for robust person reID. In particular, we have proposed a feature disentanglement method using an identity shuffling technique, which regularizes identity-related and -unrelated features and allows to factorize them without any auxiliary supervisory signals. We achieve a new state of the art on standard reID benchmarks in terms of rank-1 accuracy and mAP.

**Acknowledgments**

This research was supported by R&D program for Advanced Integrated-intelligence for Identification (AIID) through the National Research Foundation of KOREA(NRF) funded by Ministry of Science and ICT (NRF-2018M3E3A1057289).

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
