[Reviews · NeurIPS 2019]

Reviewer 1



This paper describes an approach to person re-identification that uses a generative model to effectively disentangle feature representations into identity-related and identity-unrelated aspects. The proposed technique uses an Identity-shuffling GAN (IS-GAN) that learns to reconstruct person images from paired latent representations even when the identity-specific representation is shuffled and paired with identity-unrelated representation from a different person. Experimental results are given on the main datasets in use today: CUHK03, Market-1501, and DukeMTMC. The paper is very well-written and the technical development is concise but clear. There are a *ton* of moving parts in the proposed approach, but I feel like the results would be reproducible with minimal head scratching from the written description. A few critical points: 1. The FD-GAN [27] approach is indeed very similar to the technique proposed in this submission. So much so that it is surprising (to me) that the IS-GAN outperforms it so handily. Some more analysis as to why this is the case would be useful. 2. Related to this, MGN [12] is the only approach outperforming at times IS-GAN. Deeper analysis of this and a discussion of the different affordances IS-GAN and MGN might have would help interpretation of the quantitative comparison with the state-of-the-art. 3. A grid search over six continuous parameters seems expensive. What data is this performed on? How do you set aside validation data for model evaluation? This seems to be a critical point if independent training/validation dataset isn't used for this. 4. Back on the topic of FD-GAN, the results on CUHK03 reported in [27] are significantly higher than those reported here. These FD-GAN results are omitted from Table 1, and I assume this is due to differences in experimental protocol? Some mention of this is probably warranted given the similarity of the approaches. POST REBUTTAL: The authors have addressed all of my main concerns and my decision remains to accept. I encourage the authors to include the rebuttal description of differences with FD-GAN, the hyperparameter search procedure, and the interpretation of the results with respect to WGN.

Reviewer 2



(1) Originality: This work shows a new data generation method, part-level identity shuffling technique. However, learning disentangled representation is not a new method in person reID field. ID-related or -unrelated feature learning is also similar to previous works, e.g. DG-Net ([1] Joint Discriminative and Generative Learning for Person Re-identification). ID-related or -unrelated feature is a different kind representation form of the structure and appearance features in [1]. The training strategy and most of loss functions of these two works are similar too. (2) Quality: The claim of this work can be supported by the ablation study in the experiments. The method achieves good performance on the widely-used dataset. There is no analysis of bad case, but it does not matter. (3) Clarity: Well-written. It’s easy to read. (4) Significance: This work may inspire future work to a certain extent, but it is limited to design new generation methods. The main contribution of this paper, disentangled representation, is similar to previous works.

Reviewer 3



This paper proposes a new method to learn robust representation for person re-identification problems by separating features for human identity and the others via learning generators of human images. Probably it learns GAN-based models to generate generalized human images to be robust in variations of pose and occlusion. The idea is very inspiring for applying it to other similar visual surveillance applications such as of view-point invariance or outfit-invariance. This paper is well-written and concisely focused on the main goal. However, it needs more detailed explanations for reproducibility. Specifically, the part of domain discriminators is not clear. What is the meaning of the sentence 'we add convolutional and fully connected layers for the class discriminator."? How to configure patchGAN for them? In subsection 'visual analysis for disentangled features', it would be helpful to show shuffling of two people with different colors and styles of dress as general cases for a deeper understanding of the proposed methods. I'm curious what kinds of effects are shown about camera angles, viewpoints, or outfits if possible. Here are minor comments: There are some typos and grammatical errors. It would be helpful to proofread by native speakers. In Figure 4, the box boundaries in green or red are too thin to be seen clearly. It would be better to make them thicker a little. line 26: focussed -> focused line 38: argumentation -> augmentation POST-REBUTTAL: The authors addressed all of my concerns. I've raised my score 1 higher. I am asking the authors to merge the materials of the rebuttal into the manuscript.

[Author Response · NeurIPS 2019]

# Learning Disentangled Representation for Robust Person Re-identification (ID 2853)

We thank all the reviewers for their valuable comments. We will clarify their concerns in the paper.

**FD-GAN (R1).** FD-GAN and IS-GAN are similar in that both use a GAN-based distillation technique for a robust reID. Differently, FD-GAN extracts identity-related and pose-unrelated features, but with extra pose labels. Distilling other factors except for human pose is not feasible. In contrast, IS-GAN disentangles identity-related and -unrelated features through identity shuffling, factorizing other factors irrelevant to person reID, such as pose, scale, background clutter, and occlusion, without supervisory signals for them. Accordingly, the identity-related feature for IS-GAN is much more robust to such factors of variations than the identity-related and pose-unrelated feature for FD-GAN, and this gives a superior performance on the Market-1501 and DukeMTMC-reID datasets. Note that CUHK03 was excluded, as FD-GAN used a different training/test split.

**MGN (R1).** MGN uses the same backbone network as IS-GAN to extract initial part-level features. As it is trained with a hard-triplet loss, the part-level features are highly discriminative, but they are not robust to *e.g.*, pose, scale, background clutter, and occlusion. MGN thus shows the reID performance comparable with IS-GAN on Market-1501, where discriminative attributes of identities can be captured well. For example, person images with the same identity are almost identical in the dataset. MGN, however, shows a limited performance on the CUHK03 and DukeMTMC-reID datasets, where the same person is captured with different poses, view points, background, and occlusion.

**DG-Net (R2).** DG-Net (CVPR 2019) was not published at the time of our submission. It thus should not be our consideration, but we'd like to clarify here the difference from DG-Net. Although appearance/structure features in DG-Net seem to be analogous to identity-related/-unrelated ones in IS-GAN, they are completely different. DG-Net computes appearance/structure features by AdaIN (ICCV 2017), widely used in image stylization, and thus they are more like style/content features. Figure 9 in Appendix of the DG-Net paper visualizes generated person images when structure features (analogous to identity-unrelated features of IS-GAN) are changed only. We can see that DG-Net even changes the entire attributes (*e.g.*, gender) except the color information, suggesting that structure features also contain id-related cues. Note that IS-GAN outperforms DG-Net for all benchmarks by a large margin (*e.g.*, rank-1/mAP on DukeMTMC-reID: 90.0/78.1 (IS-GAN) and 86.6/74.8 (DG-Net)).

Table 1: Quantitative comparison for a different network.

|  | $\mathcal{L}_{\mathrm{PS}}$ | R-1 | mAP |
|---|---|---|---|
|  | | Market-1501 | |
| PCB | X | 91.0 | 74.2 |
| PCB + IS-GAN | X | 92.4 | 77.2 |
| PCB + IS-GAN | ✓ | **92.7** | **77.5** |

**IS-GAN with a different backbone (R2).** To evaluate the generalization ability, we tried to use PCB as our backbone to extract CNN features, and added IS-GAN on top of the features. We modified the network architecture such that each part-level feature has the size of $1 \times 1 \times 256$ for an efficient computation, and set this as our baseline. Note that the original PCB also gives six part-level features, but with the size of $1 \times 1 \times 2,048$ for each feature. Table 1 shows that our method improves the baseline consistently, suggesting it can be applied to other methods.

**The number of body parts (R2)** We show in Table 2 the effect of the part-level shuffling loss on the different number of body parts. We can see that 1) the part-level shuffling loss generalizes well across the different number of body parts, and 2) IS-GAN shows better performance as more body parts are used.

Table 2: Ablation studies on the number of body parts.

|  | $\mathcal{L}_{\mathrm{PS}}$ | R-1 | mAP |
|---|---|---|---|
|  | | Market-1501 | |
| part-2 | X | 84.1 | 61.3 |
| part-2 | ✓ | 88.8 | 68.7 |
| part-3 | X | 86.9 | 65.7 |
| part-3 | ✓ | 91.2 | 74.2 |
| part-1,2 | X | 91.9 | 77.8 |
| part-1,2 | ✓ | 92.1 | 78.2 |
| part-1,3 | X | 93.4 | 81.1 |
| part-1,3 | ✓ | 93.7 | 81.3 |

**More results for disentangled features. (R3).** Figure 1 shows an example of generated images using a part-level identity shuffling technique. This corresponds to Fig. 5 in the main paper, but with *different identities*, demonstrating once again that IS-GAN successfully disentangles identity-related and -unrelated features in a part-level. For example, we can see, in the upper left picture, that IS-GAN changes colors of T-shirts between persons, while preserving the poses and background. On the contrary, colors of T-shirts are maintained, while the poses and background are changed in the upper right picture.

**Hyperparameter (R1).** We empirically found that training with a large value of $\lambda_{\mathrm{U}}$ is unstable. We thus set $\lambda_{\mathrm{U}}$ to 0.001 in the second stage, and increased to 0.01 in the third stage to regularize the disentanglement. We used a grid search to set other parameters with $\lambda_{\mathrm{R}} \in \{5, 10, 20\}$, $\lambda_{\mathrm{PS}} \in \{5, 10, 20\}$, and $\lambda_{\mathrm{C}} \in \{1, 2\}$ on the Market-1501 dataset. We randomly split IDs in the training dataset into 651/100 and used corresponding images as training/validation sets. Following [27, 35], we fixed $\lambda_{\mathrm{S}}$ and $\lambda_{\mathrm{D}}$ to 10 and 1, respectively. We fixed all parameters and trained our model on the CUHK03 and DukeMTMC-reID datasets.

**Discriminators (R3).** The domain and class discriminators share five blocks consisting of `conv-instnorm-lrelu`, and each has an independent head (L217-L218). For the domain discriminator, we added two more blocks, resulting in a features map of size $12 \times 4$. We then used this as an input to PatchGAN. For the class discriminator, we added one more block followed by a fully connected layer. At the time of the publication, we will make our source code and models open to the public.

Figure 1: Visualization of disentangled features for person images with different identities.

[Meta-Review · NeurIPS 2019]

All the reviewers are satisfied with the rebuttal. The authors should be strongly encouraged to integrate these discussions from the rebuttal into any final version of this work.